# Effects of Proton Irradiation on the Current Characteristics of SiN-Passivated AlGaN/GaN MIS-HEMTs Using a TMAH-Based Surface Pre-Treatment

**DOI:** 10.3390/mi12080864

**Published:** 2021-07-23

**Authors:** Young Jun Yoon, Jae Sang Lee, Jae Kwon Suk, In Man Kang, Jung Hee Lee, Eun Je Lee, Dong Seok Kim

**Affiliations:** 1Korea Multi-Purpose Accelerator Complex, Korea Atomic Energy Research Institute, Gyeongju 38180, Korea; yjyoon@kaeri.re.kr (Y.J.Y.); jslee8@kaeri.re.kr (J.S.L.); jksuk@kaeri.re.kr (J.K.S.); 2School of Electronic and Electric Engineering, Kyungpook National University, Daegu 41566, Korea; imkang@ee.knu.ac.kr (I.M.K.); jlee@ee.knu.ac.kr (J.H.L.); 3Advanced Radiation Technology Institute, Korea Atomic Energy Research Institute, Jeongeup 56212, Korea; leeeunje@kaeri.re.kr

**Keywords:** Gallium Nitride (GaN), proton irradiation, surface pre-treatment

## Abstract

This study investigated the combined effects of proton irradiation and surface pre-treatment on the current characteristics of Gallium Nitride (GaN)-based metal-insulator-semiconductor high-electron-mobility-transistors (MIS-HEMTs) to evaluate the radiation hardness involved with the Silicon Nitride (SiN) passivation/GaN cap interface. The impact of proton irradiation on the static and dynamic current characteristics of devices with and without pre-treatment were analyzed with 5 MeV proton irradiation. In terms of transfer characteristics before and after the proton irradiation, the drain current of the devices without and with pre-treatment were reduced by an increase in sheet and contact resistances after the proton irradiation. In contrast with the static current characteristics, the gate-lag characteristics of the device with pre-treatment were significantly degenerated. In the device with pre-treatment, the hydrogen passivation for surface states of the GaN cap was formed by the pre-treatment and SiN deposition processes. Since the hydrogen passivation was removed by the proton irradiation, the newly created vacancies resulted in the degeneration of gate-lag characteristics. After nine months in an ambient atmosphere, the gate-lag characteristics of the device with pre-treatment were recovered because of the hydrogen recombination. These results demonstrated that the radiation hardness of MIS-HEMTs was affected by the SiN/GaN interface quality.

## 1. Introduction

Gallium Nitride (GaN)-based devices have received attention for high-frequency and high-power applications due to their outstanding characteristics, such as low on-resistance (*R*_on_) and high speed, which can be realized by two-dimensional electron gas (2DEG), formed by a AlGaN/GaN heterostructure [1,2,3]. Moreover, high-electron mobility transistors (HEMTs) or metal-insulator-semiconductor HEMTs (MIS-HEMTs) based on the AlGaN/GaN heterojunction have been studied for electronics in the space environments because of a remarkable radiation tolerance of GaN material [4,5,6]. Electronics used in harsh space environments must be resistant to damage or malfunctions caused by ionizing radiation. In order to evaluate radiation hardness, radiation irradiation effects on the device properties have been explored in MIS-HEMTs with various gate dielectric layers, namely, SiN/Al_2_O_3_ [7,8], SiN [9,10], Gd_2_O_3_ [11], MgO/Sc_2_O_3_ [12], and poly-AlN/SiN [13]. However, these researches have focused on the impact of dielectric on radiation resistance. The impact of the radiation on the performance of GaN devices dependent on the interface between passivation and the GaN layers have not been studied yet. The pre-treatment processes before the deposition passivation layer were applied to improve the performances of GaN-based devices. In SiN-passivated devices, the pre-treatment process is based on Tetramethylammonium(TMAH) [14], NH_3_ [15], H_2_SO_4_ [16] solutions, and N_2_ plasma [17] to enhance SiN/(Al)GaN interface quality because the interface quality affects device performance and reliability. The current collapse characteristics of the AlGaN/GaN heterojunction-based devices can be especially degraded by the surface state of the (Al)GaN layer.

This study evaluated the proton irradiation effect on SiN-passivated MIS-HEMTs that use the TMAH-based pre-treatment process, which was performed to improve the SiN passivation/GaN cap interface quality. To investigate the relationship between the interface conditions and irradiation damage, we analyzed the impact of proton irradiation and the pre-treatment process on the static and dynamic current characteristics of the devices. We also verified the recovery phenomenon of the devices by re-measuring current collapse characteristics after nine months.

## 2. Device Structure and Fabrication

Figure 1a shows the schematic cross-section of the SiN-passivated AlGaN/GaN MIS-HEMT. The epitaxial structure was created using metal-orgranic chemical vapor deposition (MOCVD) equipment (SYSNEX, Korea) on a sapphire substrate. The device consisted of a 2 μm-thick GaN buffer layer, 60 nm-thick GaN channel layer, 22 nm-thick AlGaN layer, and 2 nm-thick GaN cap layer. Al composition in the AlGaN layer was 0.25. Ssheet charge density of 1.42 × 10^13^ cm^−2^ and mobility of 1330 cm^2^/V·s were identified using Hall-effect measurement at room temperature. The overall process flow for the fabrication and proton irradiation is shown in Figure 1b.

The fabrication began with the dry etching process for electrical isolation between devices. The etched depth was about 250 nm. The wet treatment process based on a TMAH solution (5% concentration) was performed at a temperature of 90 °C. The TMAH solution selectively eliminates Ga atoms on the surface as the alkaline solution and Ga-polar surface is terminated with N atoms after the treatment [18]. Thus, the TMAH-based treatment process influences the removal of native Ga-oxide and enhances the surface roughness [19]. In a previous study, we also confirmed that the TMAH treatment reduced the leakage current characteristics by effectively removing surface states on the GaN cap layer [14]. To investigate effects of the pre-treatment on the surface, we set the treatment time as 0, 1, and 3 min. We used a photoresist developer containing TMAH (AZ 300 MIF) during the photolithography steps. However, the photoresist developer had little impact on the surface because the TMAH solution reacted with the GaN material at a high temperature of ~90 °C. The TMAH in the photoresist developer was not significantly affected on the surface. We then deposited 20 nm-thick SiN as a passivation and gate dielectric layer using plasma-enhanced chemical vapor deposition (PECVD) (SINIC, Korea) at 370 °C. After the deposition of the SiN layer, source and drain contact were defined by the lithography and electron beam (e-beam) evaporator. After deposition of an Au/Ni/Al/Ti/Si multilayer, the metal layer was annealed using rapid thermal annealing (RTA) at 800 °C for 30 s in a N_2_ stmosphere. Finally, Ni/Al/Ni-based metallization was applied for the gate and pad. The current characteristics of completely fabricated devices were measured using a B1500 semiconductor device analyzer (Keysight, Santa Roda, California, USA). The gate length (*L*_G_) and gate-to-source distance (*D*_GS_) were 3 μm and 5 μm, respectively. The gate-to-drain distance (*D*_GD_) was designed to be 5, 10, 20, and 30 μm. The devices were irradiated by 5 MeV protons with a fluence of 1 × 10^14^ cm^−2^ using the RFT-30 cyclotron at the Advanced Radiation Technology Institute (ARTI). After proton irradiation, the devices were measured once again. The electrical characteristics of the devices before and after proton irradiation were compared and the effects of proton irradiation and pre-treatment on performances were analyzed. We verified an injection depth of 5 MeV protons using a simulator based on a Monte-Carlo calculation [20]. The protons with 5 MeV energy was injected up to a depth of 125 μm, generating vacancies.

## 3. Results and Discussion

Figure 2 shows transfer drain current (*I*_D_) and transconductance (*g*_m_)-gate voltage (*V*_G_) characteristics at a drain voltage (*V*_D_) of 10 V of the MIS-HEMT without and with pre-treatment for pre- and post-irradiation. The *I*_D_ and *g*_m_ characteristics of both the devices slightly decreased. 

The reduced *I*_D_ and *g*_m_ was due to the increase in sheet resistance and contact, as shown in Figure 3. The contact and sheet resistances of the devices increased after the proton irradiation because the 2DEG channel and contact were damaged by the injected protons. The proton irradiation generated defects such as Al, Ga, and N vacancies in the 2DEG channel. The defects caused a decrease in electron mobility and 2DEG sheet carrier density [21,22]. More energy loss was in the ohmic contact region due to the heavier mass of the Au atoms. The contact metal as well as the 2DEG region nearby was damaged by more scatter protons from the collisions with heavy atoms [23]. As a result, the increase in the 2DEG channel and contact resistance were caused by radiation-induced defects. A positive shift in threshold voltage (*V*_th_) was observed after the proton irradiation. The *V*_th_ was defined as the *V*_G_ intercept of the linear extrapolation of the *I*_D_ at the point of peak g_m_ (*g*_m_max_) [24], and the *V*_th_ of all the devices were extracted at the *I*_D_-*V*_G_ at a low *V*_D_ of 0.1 V. The variation rate of the *V*_th_ values before and after the proton irradiation (Δ*V*_th_) of the device without the pre-treatment was about +0.39 V. This is consistent with the results reported in Refs. [8,9]. The proton irradiation induced reduction in electron density within the 2DEG channel due to the displacement damage [25,26]. As a result, the positive shift in *V*_th_ was caused by the decreased electron density. The device with the treatment also exhibited a *V*_th_ shift of +0.35 V, as shown in Figure 2b. These results indicate that the 2DEG channel and contact resistances were degraded by the proton irradiation, irrespective of the pre-treatment process.

Figure 4a,b show the logarithmic scale *I*_D_ and *I*_G_ characteristics as a function of *V*_G_ before and after the proton irradiation of the MIS-HEMTs with and without the pre-treatment. The off-state currents (*I*_off_) of both the devices were reduced more than *I*_G._ The values of *I*_G_ were affected by the value of *I*_off_. This result indicated that the decrease in *I*_off_ was affected by the leakage current, except for the *I*_G_. The *I*_off_ of the MIS-HEMTs was formed by leakage current paths, including a buffer layer and the surface of the mesa-etched region [27,28]. 

We evaluated the buffer leakage current characteristics by measuring the current between ohmic contacts connecting the mesa-etched region, as shown in Figure 5a. In terms of buffer current characteristics before the irradiation, the structure with the treatment for 3 min was the lowest buffer current because of the enhanced SiN/GaN interface quality effects, as shown in Figure 5b. The buffer current was determined by the currents through the buffer layer and the SiN/GaN interface. A high buffer current of the structure without the treatment was induced by the leakage current path through the SiN/GaN interface, resulting from a large amount of surface states and traps related to the dangling bonds on the etched surface [29]. Because the treatment reduced these defects, the structure with the treatment obtained a relatively low buffer current. The structure with the treatment still exhibited the lowest buffer current of about 10^−9^ A/mm after the proton irradiation. The buffer currents of all structures were reduced to an almost equal rate of about 10^2^. This result is due to the defects in the buffer generated by the proton irradiation. Because protons with an identical fluence of 1×10^14^ cm^−2^ were injected into the buffer structure, the proton irradiation generated defects such as Ga vacancies, which increased the resistance of the buffer layer. Consequentially, the decrease in the buffer current led to the reduction in *I*_off_ of the MIS-HEMTs, as shown in Figure 4.

Figure 6a,b show the pulse-mode output *I*_D_-*V*_D_ characteristics before and after the proton irradiation of the MIS-HEMTs without and with the treatment. The quiescent bias of the gate and drain (*V*_G_B_, *V*_D_B_) was 0 V. We verified the output *I*_D_-*V*_D_ characteristics at the quiescent bias conditions (*V*_G_B_, *V*_D_B_ = 0 V) to minimize the bias stress and self-heating effect [30]. The pulse period and width (*P*_period_ and *P*_width_) in the pulse-mode measurement were 5 ms and 100 μs, respectively. The *I*_D_ of the device after the proton irradiation was lower than that before the proton irradiation. This result was due to a positive shift of *V*_th_ as well as an increase in sheet and contact resistances. As shown in Figure 6c, all the devices exhibited higher on-resistance (*R*_on_) after the proton irradiation than that of the devices before the proton irradiation because of the increased sheet and contact resistances. The variation of *R*_on_ as a function of *D*_GD_ exhibited a steep slope because the sheet resistance of the access region became higher [31]. This result was consistent with the transmission line method (TLM) results.

Figure 7a exhibits the current collapse characteristics of the MIS-HEMTs before and after the proton irradiation. The quiescent biases of the gate and drain (*V*_G_B_, *V*_D_B_) were (0 V, 0 V), (−10 V, 0 V), (0 V, 20 V), and (−10 V, 20 V). The pulse period and width (*P*_period_ and *P*_width_) were 5 ms and 100 μs, respectively. Before proton irradiation, the device with the pre-treatment exhibited a relatively less impact of gate bias stress on the current characteristics because of a better quality of the SiN/GaN interface. However, as the pre-treatment time increased, the gate-lag characteristics degenerated after the proton irradiation. As shown in Figure 7b, the *R*_on_ variation of the device with the pre-treatment for 3 min were significantly increased by the irradiation. The device with the pre-treatment for 3 min exhibited more damaged on the device surface with the irradiation, as shown in the microscopic image of Figure 7c. Compared to the gate-lag characteristics, the current characteristics of all the devices were hardly changed by the drain bias stress condition. These results indicated that the SiN/GaN interface was affected by the proton irradiation. The TMAH-based pre-treatment removed native Ga-oxide from the surface of the GaN cap layer and the surface became N-terminated. The adsorption of hydrogen can be enhanced by the N-terminated surface [32,33]. During the SiN deposition process, the surface was covered by the hydrogen in SiH_4_ or NH_3_ [34]. The hydrogen between the SiN/GaN interface was removed by the proton irradiation, which may generate high temperature or displacement damage. The removal of hydrogen passivation in the SiN/GaN interface forms defects and degenerates gate-lag characteristics. 

As shown in Figure 8a, MIS-HEMTs with pre-treatment for 3 min exhibited a large increase in the rate in terms of gate-lag characteristics dependent on *D*_GD_. As *D*_GD_ increased, the device exhibited a high variation in resistance because the impact of the drain bias on trapped electrons near the gate edge was reduced by a long *D*_GD_. Figure 8b shows the drain-lag characteristics of the MIS-HEMTs before and after the proton irradiation as a function of *D*_GD_. All the devices obtained a low variation rate of *R*_on_ after the proton irradiation because the *R*_on_ at *V*_G_B_ = *V*_D_B_ = 0 V was largely degraded more than *R*_on_ at *V*_G_B_ = *V*_D_B_ = 10 V. 

Figure 9a shows the *I*_D_ and *g*_m_ characteristics as a function of *V*_G_ of the device with the pre-treatment for 3 min before and after proton irradiation and nine months after the irradiation. We stored the devices in an ambient atmosphere for nine months. The degenerated *I*_D_ and *g*_m_ increased after nine months while *V*_th_ was hardly changed. Because the *g*_m_ value is associated with channel mobility [35], the current characteristics were largely recovered by the recovered mobility. The unchanged *V*_th_ means no change in the 2DEG density. The gate-lag characteristics recovered over time, as shown in Figure 9b. These results indicated that the damaged SiN/GaN interface was reconstructed by the re-passivation of hydrogen. Hydrogen was diffused from SiN or the atmosphere to the SiN/GaN interface, and formed hydrogenated vacancies. The hydrogen from SiN and the hydrogen were able to recover the proton irradiation-generated defects [36]. Thus, the performance of the devices with the pre-treatment irradiated by the protons were recovered by the hydrogen passivation effects.

## 4. Conclusions

We studied the effects of proton irradiation on SiN-passivated AlGaN/GaN MIS-HEMTs with a TMAH-based pre-treatment process for a fixed fluence of 1 × 10^14^ cm^−2^ at a proton energy of 5 MeV. The static *I*_D_ characteristics of the devices decreased regardless of the implementation of a pre-treatment process because the increase in sheet and contact resistances was caused by radiation damage. The gate-lag characteristics of the device with the pre-treatment was remarkably degenerated after proton irradiation. The hydrogen in the SiN/GaN interface formed by the TMAH-based pre-treatment and SiN deposition process was removed by the injected protons. As a result, the degeneration of the gate-lag characteristics was induced by the generated vacancies. After nine months, the current and gate-lag characteristics of the device with pre-treatment were recovered by the hydrogen re-passivation. The results of this study confirmed that the conditions of the SiN/GaN interface affected the radiation hardness of SiN-passivated MIS-HEMTs.

## Figures and Tables

**Figure 1 micromachines-12-00864-f001:**
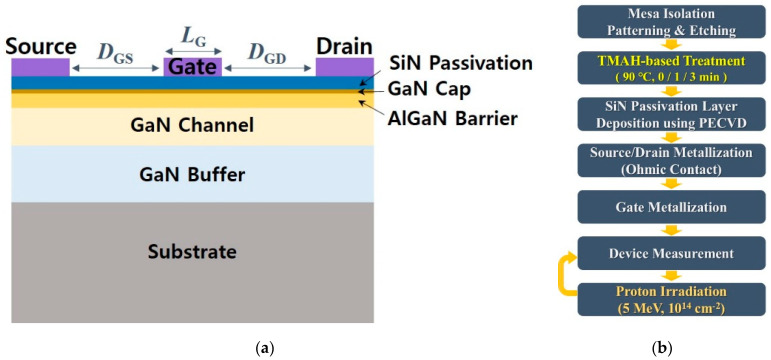
(**a**) Schematic cross-section of the SiN-passivated AlGaN/GaN metal-insulator-semiconductor high-electron mobility transistor (MIS-HEMT). (**b**) Process flow for the fabrication and proton irradiation.

**Figure 2 micromachines-12-00864-f002:**
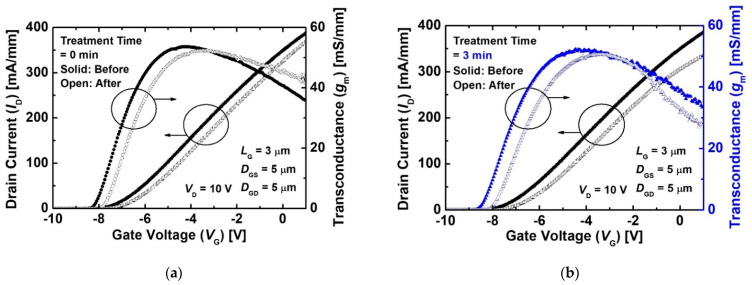
*I*_D_ and *g*_m_ characteristics as a function of *V*_G_ before and after 5 MeV proton irradiation of the MIS-HEMTs (**a**) without treatment and (**b**) with treatment for 3 min at a *V*_D_ of 10 V. The *L*_G_, *D*_GS_, and *D*_GD_ of the device were 3, 5, and 5 μm, respectively.

**Figure 3 micromachines-12-00864-f003:**
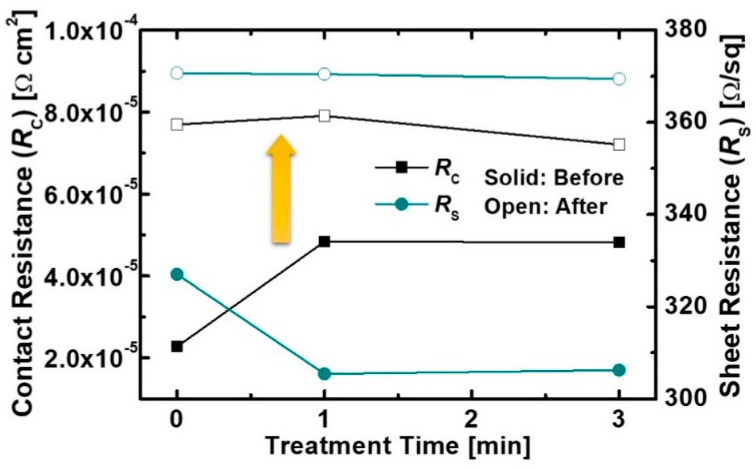
Sheet and contact resistances obtained from the transmission line method (TLM) of samples with and without treatment for 3 min before and after the proton irradiation.

**Figure 4 micromachines-12-00864-f004:**
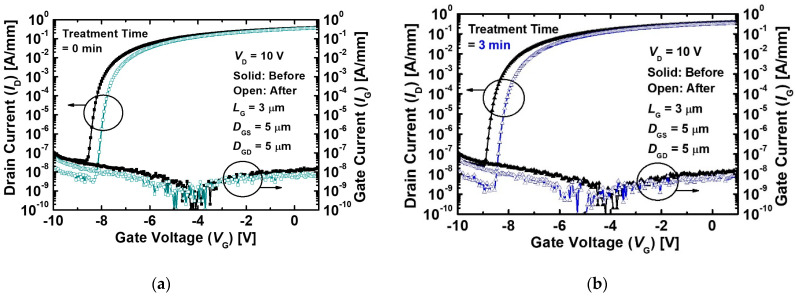
*I*_D_ and *I*_G_ characteristics as a function of *V*_G_ before and after the proton irradiation of the MIS-HEMTs (**a**) without the treatment and (**b**) with treatment for 3 min at a *V*_D_ of 10 V.

**Figure 5 micromachines-12-00864-f005:**
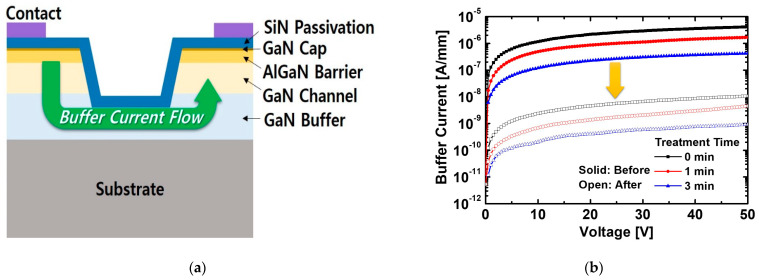
(**a**) Schematic structure for measurement of the buffer current. (**b**) Buffer current characteristics of the structure with and without treatment for 1 and 3 min before and after the proton irradiation.

**Figure 6 micromachines-12-00864-f006:**
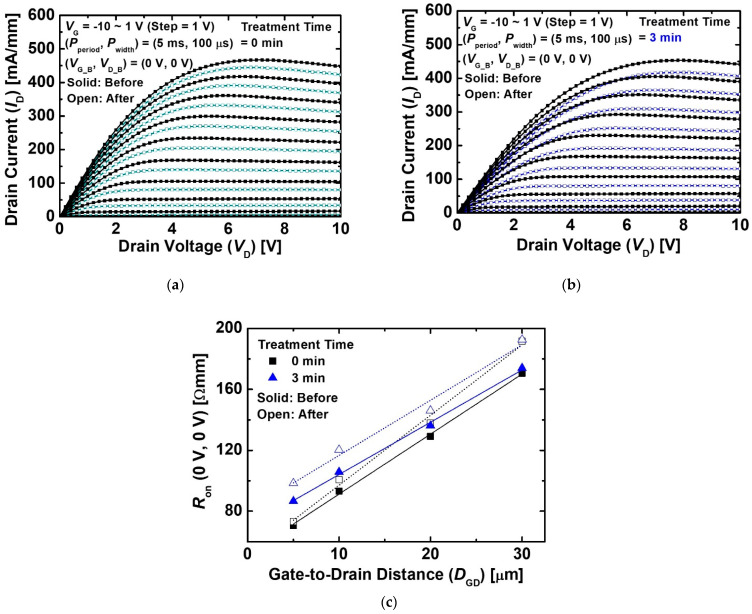
Pulse-mode output *I*_D_-*V*_D_ characteristics of the MIS-HEMTs (**a**) without treatment and (**b**) with treatment for 3 min before and after the proton irradiation. The *L*_G_, *D*_GS_, and *D*_GD_ of the device was 3, 5, and 5 μm, respectively. (**c**) Variation of *R*_on_ as a function of *D*_GD._ The *V*_G_B_ and *V*_D_B_ were 0 V. The *P*_period_ and *P*_width_ were 5 ms and 100 μs, respectively.

**Figure 7 micromachines-12-00864-f007:**
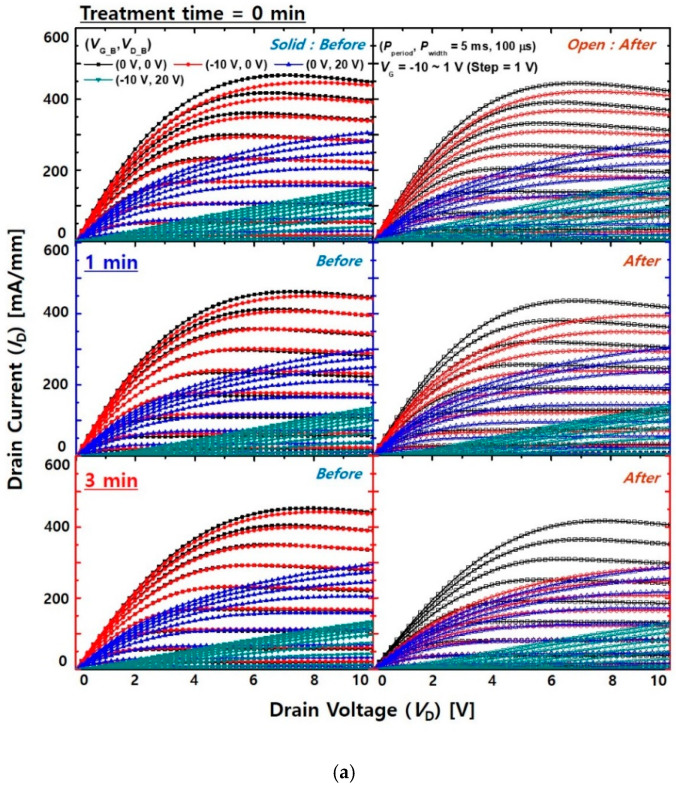
(**a**) Current collapse characteristics of the MIS-HEMTs before and after proton irradiation. (**b**) Variation rates of *R*_on_ depending on different stress bias conditions. The *V*_G_B_ and *V*_D_B_ were (0 V, 0 V), (−10 V, 0 V), (0 V, 20 V), and (−10 V, 20 V). The *P*_period_ and *P*_width_ were 5 ms and 100 μs, respectively. The *L*_G_, *D*_GS_, and *D*_GD_ of the device were 3, 5, and 5 μm, respectively. (**c**) Optical microscope image after irradiation of the devices with and without treatment.

**Figure 8 micromachines-12-00864-f008:**
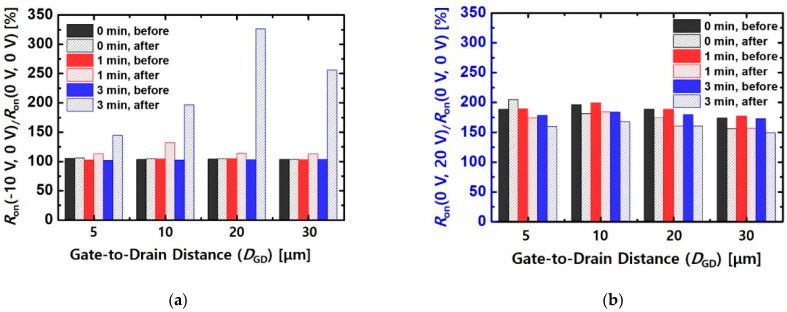
Variation in rates of *R*_on_ depending on (**a**) gate and (**b**) drain stress bias conditions of the MIS-HEMTs with different values of *D*_GD_. The *P*_period_ and *P*_width_ were 5 ms and 100 μs, respectively.

**Figure 9 micromachines-12-00864-f009:**
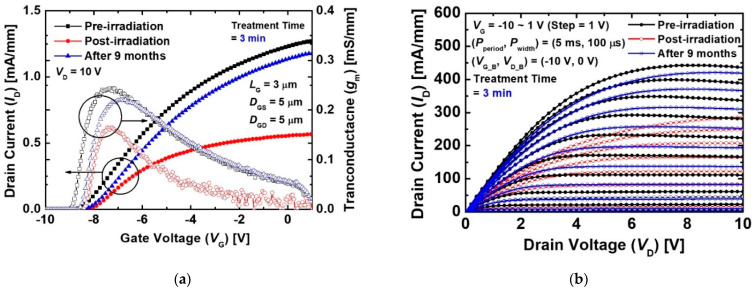
(**a**) *I*_D_ and *g*_m_ characteristics as a function of *V*_G_ at a low *V*_D_ of 0.1 V and (**b**) gate-lag characteristics of the MIS-HEMTs with treatment for 3 min before and after irradiation, and after nine months.

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
