# Peer review of "Effects of Proton Irradiation on the Current Characteristics of SiN-Passivated AlGaN/GaN MIS-HEMTs Using a TMAH-Based Surface Pre-Treatment"

_micromachines, 2021, doi:10.3390/mi12080864_

Round 1

Reviewer 1 Report

Comments on the presented manuscript.

  • the text style should be improved, some sentences are not clear, i.e. "The impact of the radiation (.....) of GaN-based devices" (lines 45-49), or "The off-state currents (....) is one of the reasons that Ioff increased" (lines 126-127). There is a lot of language errors, i.e. "...mobility (...) were identified using ...". In line 67, the "electron density" should be corrected as "sheet charge concentration" etc.;
  • there is no information what was the composition of AlGaN barrier and how the thickness of AlGaN/GaN (cap) was determined;
  • there is a thesis that irradiation causes changes in the presence of hydrogen atoms on SiN/GaN interface, however there is no appropriate measurment data which can support this.

Author Response

Reviewer(s)' Comments to Author:
Reviewer: 1
Comments to the Author
1) The text style should be improved, some sentences are not clear, i.e. "The impact of the radiation (.....) of GaN based devices" (lines 45 49), or "The off state currents (....) is one of the reasons that Ioff increased" (lines 126 127). There is a lot of l anguage errors, i.e. "...mobility (...) were identified using ...". In line 67, the "electron density" should be corrected as "sheet charge concentration" etc.;
Answer)
Thank you for kind comments. We revised the sentences in the manuscript.
“The impact o f the radiation on the performance of GaN devices dependent on the interface between passivation and GaN layers have not been studied yet. The pre treatment processes before deposition passivation layer were applied to improve the performances of GaN based devices.”
“The off state currents ( Ioff ) of both devices were reduced more than the IG. The values of IG was affected by the value of Ioff"  “A sheet charge density of 1.42×1013 cm2 and mobility of 1330 cm 2 /V·s were identified using Hall effect
measurement at room temperature.”
2) There is no information what was the composition of AlGaN barrier and how the thickness of AlGaN/GaN (cap) was determined;
Answer)
Thank you for your comments. The Al composition of AlGaN barrier is 0.25. We added the sentence for the composition of AlGaN barrier. And, w e verified the thic k ness of AlGaN/GaN layer using TEM.
 “Al composition in the AlGaN layer is 0.25" (2 th line, page 2)
3) There is a thesis that irradiation causes changes in the presence of hydrogen atoms on SiN/GaN interface, however there is no appropriate measurement data which can support this.
Answer)
We agree with your comment. Thus, w e are going to have experiment for changes in the presence of hydrogen atom. We deeply consider ed reason for the change of gate lag characteristics with various references to analyze
the degraded gate lag characteristics and recovery phenomenon. H ydrogen exists on GaN surface [R1, R2]. Since hydrogen bonding is weak, the hydrogen bonding in SiN/GaN can be broken by proton irradiation. After 9 month,
the device exhibited the recovered performance. W ithout thermal annealing, displacement damage in GaN devices can be recovered by hydrogen passi vation because many hydrogen in SiN or GaN or atmosphere exist. Thus, we
though t our opinion is reasonable . And, in order to supplement measurement date using ERD and XPS etc .., we are going to have experiment for changes in the presence of hydrogen atom as future work.
[R1] Northrup, J. E.; Neugebauer, J. Strong affinity of hydrogen for the GaN surface: Implications for molecular beam epitaxy and metalorganic chemical vapor deposion. Appl. Phys. LettAppl. Phys. Lett 20042004, 85, 3429., 85, 3429.
[R2] Bermudez, V. M. Theoretical study of hydroghydrogen adsorption on the GaN surface. en adsorption on the GaN surface. Surf. Sci. 2004, 565, 89, 565, 89--102102.

Reviewer 2 Report

1. Please quote measured Al composition in the AlGaN layer. 

2. Suggest the authors explicitly note in the manuscript that during all photolithography steps the authors did not use a photoresist developer containing TMAH as this would result in additional exposure times of the surface to TMAH. Does the photoresist developer used affect the surface in any way?

3. Editorial comment: Page 5 the sentence "The structure with the treatment 143 still exhibited the lowest buffer current of about 109 A/mm after the proton irradiation." There should be a negative sign in the exponent.

4. Editorial comment: In the introduction the sentence "The 45 impact of the radiation on the performance of GaN devices dependent on the interface 46 between passivation and GaN layers have not been studied yet although the pre-treatment processes before deposition passivation layer were applied to improve the performances of GaN-based devices." Suggest breaking into two sentences.

5. The authors mention that the proton irradiation leads to point defect generation which in turn affects things like contact and sheet resistance and thereby device performance. Suggest the authors spend more time explaining the likely point defects being generated (with citations). Future work it would be interesting if the authors could measure these vacancies directly - if possible.

6. For future work, it would be interesting if the authors could do a similar study as a function of fluence and quantify any changes in vacancy formation.

Author Response

Comments to the Author
1) Please quote measured Al composition in the AlGaN layer.
Answer)
Thank you for your comments. We added the sentence for Al composition in the AlGaN layer.
"Al composition in the AlGaN layer is 0.25.” (2th line, page 2)
2) Suggest the authors explicitly note in the manuscript that during all photolithography steps the authors did not use a photoresist developer containing TMAH as this would result in additional exposure times of the surface to TMAH. Does the photoresist dev eloper used affect the surface in any way?
Answer)
We used a photoresist developer containing TMAH (AZ 300 MIF). TMAH s can react with GaN material at a high temperature of ~ 90 ℃. Thus, the photoresist developer have little impact on the surface. We added the sentences
for effect of the used photoresist developer on surface.
"We used photoresist developer containing TMAH( AZ 300 MIF) during photolithography steps. Photoresist developer have little impact on the surface because the TMAH solution react with the GaN material at a high temperature of ~ 90 ℃. The TMAH in photoresist developer was not significantly affected on the surface. (3rd line-6th line, page 3)
3) Editorial comment: Page 5 the sentence "The structure with the treatment still exhibited the lowest buffer current of about 10 9 A/mm after the proton irradiation." There should be a negative sign in the exponent.
Answer)
Thank you for kind comments. We added a negative sign in the exponent. (109 A/mm -> 10-9 A/mm) (11th line, page 5)
4) Editorial comment: In the introduction the sentence "The impact of the radiation on the performance of GaN device s dependent on the interface between passivation and GaN layers have not been studied yet although the
pre treatment processes before deposition passivation layer were applied to improve the performances of GaN based devices." Suggest breaking into two sentences.
Answer)
We revised the sentence considering the reviewer comment
 “The impact of the radiation on the performance of GaN devices dependent on the interface between passivation and GaN layers have not been studied yet. The pretreatment processes treatment processes before deposition passivation layer were applied to improve the performances of GaN passivation layer were applied to improve the performances of GaN-based devices. "
(1st line- 5th line, page , page 22)
5) The authors mention that the proton irradiation leads to point defect generation which in turn affects things like The authors mention that the proton irradiation leads to point defect generation which in turn affects things like contact and sheet resistance and thereby device performance. Suggest the authors spend more time explaining the likely point defects being likely point defects being generated (with citations). Future work it would be interesting if the authors could measure these vacancies directly measure these vacancies directly -- if possible.
Answer)
We added the sentences about the defect generation and impact of defect on resistance. And, we agree with comments. Thus, in future work, we are going to measure these vacancies. 
“The proton irradiation generated defects such as Al, Ga, and N vacancies in the 2DEG channel. The defects caused the decrease in electron mobility and 2DEG sheet carrier density [21, 22]. More energy loss was in ohmic contact region more due to heavier mass of Au atoms. The contact metal as well as the 2DEG region nearby contact was damaged by more scattering protons from the collisions with heavy atoms [23]. As a result, the increase in 2DEG channel and contact resistance were caused by radiation-induced induced defects.” (30th line, page 3, 3rd line, page 4)
[21] Polyakov, A. Y.; Pearton, S. J.; Frenzer, P.; Ren, F.; Liu, L.; Kim, J. Radiation effects in GaN materials and devices. J. Mater. Chem. C2013, , 1, 8771, 877-887.
[22] Auret, F. D.; Goodman, S. A.; Koschnick, F. K.; Spaeth, J.--M.; Beaumont, B.; Gibart, P. Proton bombardmentbombardment--induced electron traps in epitaxially grown induced electron traps in epitaxially grown n-GaN. GaN. Appl. Phys. Lett. 1999, 74, 407-409.
[23] Liu, L.; Cuervo, C. V.; Xi, Y.; Ren, F.Liu, L.; Cuervo, C. V.; Xi, Y.; Ren, F.; Pearton, S. J.; Kim, H.-Y.; Kim, J.; Kravchenko, I. I. Impact of proton irradiation on dc performance of AlGaN/GaN high electron mobility transistors. of proton irradiation on dc performance of AlGaN/GaN high electron mobility transistors. J. Vac. Sci. J. Vac. Sci. Technol. B  2013, 31, 042202. (page 10)
6) For future work, it would be interesting if the For future work, it would be interesting if the authors could do a similar study as a function of fluence and any changes in vacancy formation quantify any changes in vacancy formation.
Answer)
Thank you for kind comments. We are going to have additional experiment for radiation test as a function of fluence and to quantify vacancy formation.
